# The Pathopharmacological Interplay between Vanadium and Iron in Parkinson’s Disease Models

**DOI:** 10.3390/ijms21186719

**Published:** 2020-09-14

**Authors:** Samuel Ohiomokhare, Francis Olaolorun, Amany Ladagu, Funmilayo Olopade, Melanie-Jayne R. Howes, Edward Okello, James Olopade, Paul L. Chazot

**Affiliations:** 1Department of Biosciences, Durham University, County Durham DH1 3LE, UK; samokhare@yahoo.com (S.O.); afolaolorun@outlook.com (F.O.); amanykladagu@ymail.com (A.L.); 2Department of Veterinary Anatomy, University of Ibadan, Ibadan 200284, Nigeria; jkayodeolopade@yahoo.com; 3Department of Anatomy, College of Medicine, University of Ibadan, Ibadan 200284, Nigeria; fe.olopade@mail.ui.edu.ng; 4Natural Capital and Plant Health Department, Royal Botanic Gardens Kew, Surrey TW9 3DS, UK; M.Howes@kew.org; 5Human Nutrition Research Centre, Population Health Sciences Institute, Faculty of Medical Sciences, Newcastle University, Tyne and Wear NE1 7RU, UK; edward.okello@newcastle.ac.uk

**Keywords:** vanadium, iron, mitochondria, oxidative stress, motor activity, Parkinson’s, RONS

## Abstract

Parkinson’s disease (PD) pathology is characterised by distinct types of cellular defects, notably associated with oxidative damage and mitochondria dysfunction, leading to the selective loss of dopaminergic neurons in the brain’s substantia nigra pars compacta (SNpc). Exposure to some environmental toxicants and heavy metals has been associated with PD pathogenesis. Raised iron levels have also been consistently observed in the nigrostriatal pathway of PD cases. This study explored, for the first time, the effects of an exogenous environmental heavy metal (vanadium) and its interaction with iron, focusing on the subtoxic effects of these metals on PD-like oxidative stress phenotypes in Catecholaminergic a-differentiated (CAD) cells and PTEN-induced kinase 1 (PINK−1)^B9^
*Drosophila melanogaster* models of PD. We found that undifferentiated CAD cells were more susceptible to vanadium exposure than differentiated cells, and this susceptibility was modulated by iron. In PINK−1 flies, the exposure to chronic low doses of vanadium exacerbated the existing motor deficits, reduced survival, and increased the production of reactive oxygen species (ROS). Both *Aloysia citrodora* Paláu, a natural iron chelator, and Deferoxamine Mesylate (DFO), a synthetic iron chelator, significantly protected against the PD-like phenotypes in both models. These results favour the case for iron-chelation therapy as a viable option for the symptomatic treatment of PD.

## 1. Introduction

Parkinson’s disease (PD) is the second-most prevalent neurodegenerative disease, after Alzheimer’s disease, and affects from one to five percent of the world’s elderly population, i.e., over 60 years to over 85 years, respectively [1]. Its prevalence has steadily increased in recent times, and this has been proposed to be associated with exposure to environmental toxicants, including heavy metals. Indeed, metal dysregulation with a resultant neurodegenerative disease is now of great public health concern [2,3]. The pathogenesis of PD remains unclear, being a combination of genetic mutations and environmental factors. However, it is thought to be linked more to environmental and epigenetic factors rather than purely genetic factors, as over 90% of PD cases are of sporadic nature, whilst approximately 5−10% are of familial origin, caused by gene mutations [4]. Single-gene mutations leading to PD can be divided into two types: Autosomal Dominant Parkinson, including the α-synuclein non-A4 component of the amyloid precursor (NAC/α-synuclein) and leucine repeat rich kinase 2 (LRRK2) [5], and Autosomal Recessive Parkinson (AR-JP), including PARKIN, DJ-1, P type ATPase (ATP13A2), and (PTEN)-induced kinase 1 (PINK-1).

Several heavy metals are required for vital physiological processes in the body, including the brain. Useful metals in the brain include copper, a vital cofactor for enzymes in neurotransmitter synthesis; zinc, which is needed in synaptic transmissions and regulates NMDA glutamate receptor activity; iron, for systemic oxygen transport, myelin, and neurotransmitter synthesis; and manganese, a cofactor in metalloproteins and associated with enzyme functions, including hydrolases and lyases [6,7,8,9]. In contrast, occupational and chronic environmental exposures to metals, especially copper, zinc, lead, and iron, have long been reported as risk factors for neurodegenerative diseases, including PD [10,11,12]. The exposure to low doses of certain metals, cumulatively, results in the accumulation of these metals in the brain. A study of metal concentration and distribution in the human olfactory bulb (the first port of entry for airborne environmental toxicants) revealed elevated levels of copper, zinc, and iron corresponding to areas of α-synuclein aggregation, perhaps relevant to the premotor anosmia symptom in PD [13]. The substantia nigra, which is the main brain region affected in the pathogenesis of PD, has long been reported to be susceptible to metal toxicity. High levels of aluminium, zinc, and iron have been found in the substantia nigra of PD patients, compared to those of matched controls [14,15,16]. Iron chelation has been proposed as a disease-modifying strategy for PD, with the iron chelator drug deferiprone currently being investigated clinically in Parkinson’s patients on this basis (https://clinicaltrials.gov/ct2/show/NCT02655315; https://clinicaltrials.gov/ct2/show/NCT02728843).

Vanadium (VD) is the fifth-most abundant transition metal in the Earth’s crust [17]. It is known to be an essential trace element for plants and animals; it stimulates chlorophyll synthesis in plants and the growth of young animals, but it does not yet have any essential role established in humans. The addition of VD to feeds was reported to increase the growth of rats by 40% [18]; goats exposed to VD-deficient feeds had reduced growth, reproductive performance, and life expectancy [19]; and rats with VD-deficient diets had reduced thyroid weights and hormone functions [20]. Nielsen [21] opined that VD mimics growth factors like the epidermal growth factor, fibroblast growth factor, or insulin, thus stimulating bone cell proliferation and collagen synthesis. VD is released into the Earth’s crust through the mechanical and chemical weathering of rocks, volcanic emissions, and forest fires. However, VD, as a trace contaminant of coal and crude oil, is released into the atmosphere during their combustion, as well as during the mining of VD-containing ores [22]. These VD emissions in the air also find their way into the soil, surface, and deep-water systems. This pollution of the natural systems has been on the increase over the last few decades. VD is widely used in steel-manufacturing processes, such as building planes and ships, the production of steel alloys of high tensile strength, and paint and pigment manufacturing, as well as the production of redox batteries [22,23]. It is also typically present in welding fumes as vanadium pentoxide (V_2_O_5_) and, thus, is emitted by welding rods commonly used in construction.

Low-dose VD is known to have some health benefits, notably blood sugar-lowering properties in diabetic patients and supplements in leukaemia treatments [24]. However, VD exposure has been recently implicated in the promotion of apoptotic cell death in substantia nigral dopaminergic neurones via caspase-3-dependent protein kinase C delta (PKCδ) cleavage [25], as well as dopamine depletion and the severe loss of dopaminergic neurones in the olfactory bulb [26]. These effects have implications for PD early pathogenesis and development. The PD pathology is characterised by distinct types of cellular defects: abnormal protein aggregation, and oxidative damage, which relates to mitochondrial dysfunction and a selective loss of dopaminergic neurons in the substantia nigra pars compacta (SNpc).

Iron and VD are both transition metals, which are known to cause oxidative stress by the generation of superoxide and hydroxyl radicals through the Fenton reaction. This leads to free radical-mediated damage to proteins, membranes, and DNA [27]. Vanadium induces mitochondrial stress by causing a gradual opening of the permeability transition pore (PTP), thus leading to mitochondrial swelling, collapse of the mitochondrial membrane potential, and release of cytochrome c and culminating in mitochondrial-mediated cell apoptosis [28,29,30]. The relationship between iron and vanadium in the body system is complex and not fully understood. However, the following have been reported: similar to iron, transferrin binds about 90% of the vanadium in plasma, but it is not yet established if its release is mediated through ferritin [31]. Studies have, however, indicated that the iron stores in ferritin may be released by vanadium present in the system. Monteiro et al. [32] reported that vanadyl ions, but not vanadate, caused iron release from ferritin stores, while other studies have suggested that vanadium induced ferritin molecule degradation, thus leading to an iron-mediated free radicals generation in astrocytoma cells [33]. This suggestion is further strengthened by the fact that many studies have reported an amelioration of vanadium-mediated toxicity when iron chelators, such as deferoxamine (DFO), are used [34]. Although the neurotoxicological effects of vanadium have been described in some detail, the possible subtoxic pharmacological, cell biological, or functional effects have yet to be explored. Furthermore, in the brains of PD patients, the SNpc has also been found to have significantly higher levels of iron than age-matched controls, and this elevated iron has been associated with mitochondrial dysfunction, leading to endoplasmic reticulum (ER)- and oxidative stress. Although a close interaction between VD and iron has been recently reported in glial cells, limited studies have been performed in neurons.

The aim of this study was to establish the effects of the exogenous environmental heavy metal VD on the neuronal mitochondrial viability and oxidative status, using a monoaminergic CAD (Cath. a-differentiated) cell line in vitro, and, thence, the motor activity, brain oxidative status, and life span, using the in vivo PINK-1^B9^ model of PD. The interplay between VD and endogenous iron was, consequently, explored in both models.

## 2. Results

### 2.1. Cell Studies

#### Dose-Dependent Effect of Vanadium on Undifferentiated and Differentiated CAD Monoamine Neuronal Cells, Evaluation of Their Iron Content, and Effect of Iron Chelation

The first set of experiments were designed to explore the dose-dependent effects of vanadium (VD) on a range of immature and mature monoamine neurons in the culture. Acute (one-day) and chronic (five-day) treatments with graduated doses (10–1000 μM) of sodium metavanadate (vanadium) were performed for undifferentiated and differentiated cells. The mitochondrial viability investigation revealed that undifferentiated cells were more sensitive to the toxic effects of both acute (from 100 μM) and chronic (>20 μM) administrations of VD, but differentiated cells were only affected by chronic administration (from 100 μM) (Figure 1A). Both differentiating and differentiated cells were insensitive to acute VD administration up to 1000 µM (Appendix A).

The levels of iron in undifferentiated and differentiated cells were then determined with a view to establish if iron levels at different stages of development may account for the variations in sensitivity of CAD cells to vanadium exposure. We thus established that the increased sensitivity of the undifferentiated cells was associated with significantly higher (*p* < 0.005) levels of intracellular iron (about a three-fold increase) in undifferentiated versus differentiated CAD cells (Figure 1B).

The effects of both synthetic (DFO) and natural iron chelators (*Aloysia citrodora* essential oil) upon vanadium (200 μM)-induced oxidative stress (mitochondrial viability) following chronic (six-day) exposure in differentiating CAD cells was investigated. Both DFO and *A. citrodora* oil significantly reversed vanadium-induced toxicity, as compared with vanadium-only treated cells (Figure 2).

### 2.2. Fly Studies

#### 2.2.1. Time-Dependent Effect of Chronic Exposure of Subtoxic Vanadium on the Motor Activity and Survival of Wild-Type (WT) and PINK-1 Mutant Flies

A progressive decrease in the motor activity of both types of flies with increasing age was observed. PINK-1 flies showed a highly significant lower motor activity than the WT flies (Figure 3A,C). In the WT flies, chronic exposure to subtoxic doses of VD revealed no significant effect (*p* > 0.05); however, a modest increase in motor activity was seen in flies treated with L-dopa (* *p* < 0.05) compared to the control group (Figure 3B). In contrast, VD significantly exacerbated the existing locomotor deficits in mutant PINK-1 flies (** *p* < 0.01), while L-dopa ameliorated them (Figure 3D).

Like the effect on motor activity, chronic VD had no general effect on the survival of wild-type flies apart from a modest effect seen in the first week (day 5). However, a significant increase in the survival of flies treated with L-dopa (** *p* = 0.0016) was observed (Figure 4A). In PINK-1 mutant flies, a reduction in survival in the VD-treated group (* *p* = 0.0349) relative to the control group was observed, with the median survival for the VD-treated group being five days compared to eight days for control flies. There was no significant effect on survival in the L-dopa-treated group relative to the control, with the median survival for both groups being eight days (Figure 4B).

#### 2.2.2. Effect of Iron Chelation on Motor Activity and ROS Generation

To determine the effect of iron chelation on the VD-exacerbated motor deficits in PINK-1 mutant flies, DFO was given to a subset of them. A significant improvement was observed in the motor activity in the presence of the iron chelator (DFO) + VD (**** *p* < 0.0001) compared to VD alone and to control PINK-1 mutant flies (Figure 5A). Concomitant measurements of ROS production in PINK-1 mutant fly brains revealed that treatments with low doses of VD enhanced ROS generation after 14 days, compared with the control group, and that iron chelation (DFO) significantly reversed this (Figure 5B). Conversely, in the WT flies, treatments with low doses of VD significantly reduced ROS generation after 14 days compared to the controls, and iron chelation (DFO) completely reversed this VD-induced reduction of ROS (Appendix A).

Comparing the total thiol (T-SH) levels in the wild-type and PINK-1 mutant control flies, we observed a surprisingly high T-SH baseline level in the mutant compared to the wild-type flies. Chronic low-dose VD (1μM) elicited a reduction in T-SH in WT flies but no significant effect upon PINK-1 mutant flies. Iron chelation (DFO) reversed the effect of VD on TS-H levels in the WT flies to the control levels (* *p* < 0.05) but significantly reduced the T-SH levels in PINK-1 mutant flies (* *p* <0.05), compared to the control and VD-treated PINK-1 mutant flies (Figure 6).

## 3. Discussion

Oxidative stress plays a prominent role in vanadium (VD) toxicity and Parkinson’s disease (PD). The ability of VD to generate reactive oxygen species (ROS) in a Fenton-like reaction has been reported [35], but there is strong evidence to suggest that VD also generates ROS indirectly by releasing iron from intracellular stores [36]. In this present study, CAD monoaminergic neuronal cells were shown to contain higher iron contents in immature undifferentiated cells in comparison to mature fully differentiated cells. This correlated well with the higher susceptibility to VD in undifferentiated versus differentiated cells. Furthermore, the chronic treatments of undifferentiated, differentiating, and differentiated cells displayed time- and dose-dependent sensitivity to VD (Appendix A). In agreement with this initial observation, treatments of the differentiated cells with both the synthetic iron chelator (DFO) and the natural iron chelator, *Aloysia citrodora* essential oil [37], completely inhibited the oxidative stress-induced deleterious effects of VD. This is in close agreement with a recent report that showed that rats raised on an iron-deficient diet were resistant to a VD insult [38]. Furthermore, the presence of high levels of iron stores in glia cells, e.g., oligodendrocyte progenitor cells (OPCs) versus mature oligodendrocytes and astrocytomas versus normal astrocytes, has been previously suggested to be key in their vulnerability to VD toxicity [33,36].

The PTEN-induced kinase 1 (PINK-1) gene encodes a kinase located on the membrane of the mitochondrion [39], and the mutation of PINK-1 leads to brain mitochondrial dysfunction and the heightened susceptibility of neurons to oxidative stress. The loss of PINK-1 causes an inhibition/reduction of the mitochondrial respiratory capacity, with Complexes I and II being the most affected [40]. This was, however, found to have a predilection for striatal neurones, especially dopaminergic neurones, which are known to have a high-energy demand [41]. Interestingly, the mitochondrion is also proposed as an important target for vanadium accumulation [42], which has been reported to induce mitochondrial membrane depolarization, inhibit mitochondrial oxygen consumption, and depolymerise heat shock protein 60 (Hsp60), a key mitochondrial chaperone protein [28,43]. The mitochondrion is also the site where iron is transformed into its bioactive form by the heme and iron–sulphur cluster (ISC) [44]. Once iron enters into the cell, most of it goes to the mitochondria in order to support the many iron-dependent functions of the cell, including those that require ISC and/or heme prosthetic groups, to support enzymatic or structural functions. Iron homeostasis is, thus, regulated by the efficiency of the mitochondrial heme and Fe/S protein assembly [45].

The next stage of the study explored the effects of a subtoxic concentration of VD and the role of iron upon wild-type and PINK-1 mutant loss-of-function flies as a model of familial Parkinson’s disease with a mitochondrial respiratory exchange deficit. In wild-type flies, the dopamine precursor L-DOPA, which rescues dopaminergic neurones in cellular stress, elicited a significant increase in both the motor activity and survival, while vanadium had no significant effect (positive or negative).

VD was shown to significantly reduce and increase ROS levels in WT and PINK-1 mutant flies, respectively. Thiols (e.g., glutathione) are prominent components of the body’s antioxidant defence system that aid the protection of proteins from ongoing irreversible oxidative damage [46]. The total thiol (T-SH) level in the body normally reflects the oxidative stress status of the body; thus, chronic diseases in which oxidative stress is implicated are usually associated with depleted total thiol levels. In fact, [47] recently reported a significant depletion in total glutathione (GSH-T) in the PINK-1^B9^
*Drosophila* model of PD. This is in contrast to our findings, but it should be noted that the flies in this present study were raised on a different genetic background. Furthermore, our study measured the total thiol levels rather than the GSH-T. In certain situations, there can be an elevated thiol level in response to acute oxidative stress, as seen in a report of 24-h exposure to paraquat in *Drosophila* flies [48]. Our study showed a clear significant increase in the total thiol level of the PINK-1 mutant flies relative to the wild type (Figure 6). Though we expected to find a reduction in T-SH levels in the PINK-1 flies, as one would expect in an experimental model of a chronic disease with an oxidative stress component, such as PD, we did not find (at the time of writing) any previous report in the literature on the T-SH level in PINK-1 mutant flies. Therefore, there is, to our knowledge, no precedent to compare our findings.

Low-dose subtoxic VD administration in normal cells leads to a promotion of cellular wellbeing, possibly through the modulation of growth factor-mediated signal transduction pathways and the promotion of cell transformation [49]. In cell culture studies, low-dose VD (1 µM) led to the proliferation of astrocytes and mature and immature oligodendrocytes, which was attributed to the above mechanisms [29]. In the PINK-1 mutation, however, this beneficial effect is lost, thus making the cell more susceptible to the deleterious effects of ROS. Interestingly, VD had no significant effect upon the apparently maximal T-SH levels in PINK-1 flies.

To explore the role of iron on the effects of VD, the iron chelator DFO was employed again. Coadministration of the DFO with VD in WT flies restored the reduced ROS and thiols levels caused by VD alone back to the control levels (Appendix A). PINK-1 mutant flies displayed a clear deficit in motor activity in parallel with raised ROS levels, compared to WT, as reported widely [50,51] in this model. Notably, in PINK-1 mutant flies, subtoxic VD chronic administration exacerbated both motor activity deficits and reduced survival levels. At this time point, chronic iron chelation with DFO provided a significant improvement in the motor activity, completely reversed the enhanced ROS generation, and partially reduced the apparently maximal T-SH levels in PINK-1 flies exposed to low-dose VD, indicating a partial iron-dependent oxidative stress mechanism.

Both iron and VD are known to bind to transferrin in the circulation [31], but the most important protein in iron homeostasis is ferritin. Once released, ferritin iron can participate in free-radical reactions, thus resulting in oxidative damage [52]. Several xenobiotics have been implicated in causing this iron release, e.g., paraquat, adriamycin, and alloxan, leading to oxidative stress and lipid peroxidation [53,54]. Monteiro et al. [32] reported a similar release of iron stores from ferritin by vanadyl in a concentration-dependent fashion, thus stimulating a vanadium-dependent lipid peroxidation. Clearly, cells with higher stores of iron would be more sensitive to the presence of VD, as it dislodges intracellular iron to increase the oxidative stress, damage, and apoptosis. This susceptibility was, therefore, found to be ameliorated by the administration of DFO in this study.

The function of iron chelators in mopping up excess iron ions would make them useful in mitigating against effects observed from the “iron-mediated oxidative damage” caused by VD administration, as observed in this study. This is further illustrated by [38], who reported that VD induced OPCs depletion, hypomyelination, and neurobehavioral deficits, which were partially protected by iron deficiency. This lends credence to the theory that vanadium can indirectly cause oxidative damage by releasing iron from intracellular stores. Both DFO and *A. citrodora* are known iron chelators that mop up excess iron in the system, both in vivo and in vitro. However, no chelator is entirely specific for one given metal; both DFO and deferiprone are known to chelate metals other than iron (3^+^) [55]. In fact, DFO, Tiron, DFOA (Deferoxamine Mesylate), and DTPA have been reported to chelate VD—particularly, Van (4^+^) in vivo [56,57,58]. Tubafard et al. [59] reported an inverse relationship between the cellular iron levels and exogenously administered VD. After chelation with DFO, VD levels in the tissues were significantly reduced, while iron levels simultaneously returned to normal, and the symptoms of toxicity were also reduced. Our use of chelators, thus, ameliorated the oxidizing effect of VD on PINK-1 flies. Despite antioxidants having positive results in experimental exposure to VD [58,59], we can speculate from our findings that a product with an iron chelating property in addition to an antioxidant property will be more desirable as a neuroprotectant. This provides new evidence that iron/vanadium chelation may be valuable in the management of neurological diseases, including PD.

## 4. Materials and Methods

### 4.1. In Vitro Studies

The monoaminergic CAD (Cath. a-differentiated) cell line (Biosciences Department, Durham University, Durham, UK) exhibit biochemical and morphological characteristics of mature differentiated primary neurons, expressed enzymatically active tyrosine hydroxylase, and accumulated L-dopa, the precursor of dopamine.

### 4.2. Aloysia Citrodora Palau Essential Oil

Chemically characterised *Aloysia citrodora* Paláu essential oil, determined by gas chromatography-mass spectrometry (GC-MS), as described previously (see Abuhamdah et al., 2015), was utilised for this study.

#### 4.2.1. CAD Cell Differentiation

CAD cells were grown at 37 °C and in 5% CO_2_ on 75-cm^2^ tissue culture flasks (Sarstedt, Newton, NC, USA) in Dulbecco’s modified Eagle’s medium DMEM⁄F-12 Media-GlutaMAX™-I (GIBCO, Grand Island, NY, USA), supplemented with 10% foetal bovine serum (FBS; Sigma, St. Louis, MO, USA). Cells were passaged every 3 to 4 days at a 1:4 dilution. To differentiate CAD cells, the growth media (with 10% FBS) in which the cells had been plated was removed gently and replaced with serum-free DMEM Nutrient Mixture F-12. After replacement of the serum-free media, the cells were returned to the incubator and left for six days to differentiate (DIV6). Differentiation was confirmed by microscopic analysis and MAP2 immunolabelling (not shown).

#### 4.2.2. MTT Cell Proliferation (Mitochondrial Viability) Assay

To determine the cell mitochondrial viability, the MTT (3,-4, 5 dimethyithiazol-2, 5 diphenyl tetrazolium bromide) assay was performed as previously described [60,61]. The formazan crystal optical density was spectrophotometrically read at 595 nm. (Thermo Lab Systems Multiskan Ascent, V1.3, MA, USA).

#### 4.2.3. Cell Culture Iron Metal Determination

CAD cell cultures and experiments were performed in Containment Level 2 facilities under sterile conditions. CAD cells were grown at 37 °C and in 5% CO_2_ on 75-cm^2^ tissue culture flasks (Sarstedt, Newton, NC, USA) in Dulbecco’s modified Eagle’s medium DMEM⁄F-12 Media-GlutaMAX™-I (GIBCO, Grand Island, NY, USA), supplemented with 10% FBS. CAD cells were passaged and seeded as previously described into two 75-mL flasks. After 24 h, the process of differentiation was initiated. After 100% confluency was achieved, the undifferentiated cells were rinsed with cold PBS and detached. After detaching, the falcon tubes were centrifuged for 5′ (4 °C, 800× *g*) and transferred into 1.5-mL Eppendorf tubes, re-centrifuged, and stored in −20 °C for Inductively coupled plasma mass spectrometry (ICP-MS) analysis for metal contents. Similarly, the differentiated (6 days DIV) cells were detached and stored in −20 °C for ICP-MS.

#### 4.2.4. ICP-MS Analysis

One point five millilitres of nitric acid (HNO_3_, 65%) was added to each thawed cell pellet and vortexed. The cells were left for 24 h to be digested in the fume hood and vortex-mixed again. One point two millilitres of digested sample was removed and dispensed into fresh Eppendorf tubes and centrifuged (5 min, 4000 rpm). Five hundred microlitres of a digest was then diluted with 4.5-mL 2.5% HNO_3_. One millilitre of the clarified sample was decanted into a 15-mL falcon tube. The clarified sample was diluted 1/10 (2% HNO_3_). The analysis was carried out with Ag as an internal standard (Ag 100 ppb), which was added to the samples and standard curve. An ICP-MS analysis was performed in duplicates.

Standard solutions of known metal concentrations were analysed before and after the samples (“front standard curve” and “back standard curve”), and these were used to define the limits for quantification. The final working range was used to quantify the metal content.

### 4.3. In Vivo Model—Drosophila Melanogaster Flies

#### 4.3.1. Drosophila Melanogaster Fly Stocks and Culturing Conditions

*Drosophila melanogaster* wild-type Dahomey (WT) and PINK-1 (w-PINK-1 B9/FM7.GFPw+) mutant flies were used for this experiment. Flies were maintained in an incubator with a 12-h day-night cycle at 25 °C. Fly food was prepared by mixing 15.025 g of instant medium (Jazz-Mix Drosophila Food, Thermo Fisher Scientific, MA, USA) with 46 mL of deionized water (formula with this food: water ratio was found to have the best consistency) per bottle (Okello and Tan, personal communication). Following this, each bottle was shaken to ensure even distribution of the water—after which, a few grains of baker’s yeast were added to each bottle. Each bottle was plugged with a foam plug and left to set for at least an hour at room temperature before transferring the flies (progenies) into them. Fresh food was prepared every two weeks, and flies were flipped into new bottles every 2 to 3 days.

#### 4.3.2. Dosage and Treatment

This study explored the effects of chronic exposure of VD on the motor activity, lifespan, and oxidative stress markers (5 replicates of *n* = 10 per bottle per group). Treatments were commenced from the larval stage and continued throughout the experiments. The groups (male flies) tested comprised: a control group (with no treatment), positive control (treated with L-dopa 13.4 µM), a treatment group (VD, 1 µM only), and another treatment group (VD + DFO (10 µM)) to investigate the influence of iron chelation. To prevent the oxidation of L-Dopa, 1.6 µM of ascorbic acid was added to each bottle containing L-Dopa [62]. At the start of each experiment, flies were allowed to lay eggs on food, and once the larvae appeared, the adult flies were released from the bottles, and after 6 days, the progenies were transferred to fresh food and redistributed onto treatments the next day (which was counted as test day zero). Male flies were selected for the experiment. To establish the effect of synthetic iron chelation on the motor activity and lifespan, a dose-dependent effect of DFO (0−20 µM) was tested in both WT type and PINK-1 flies (not shown). From the results of this experiment, we established the optimal concentration of synthetic iron chelator deferoxamine (DFO) mesylate salt required for subsequent experiments as 5 µM for WT and 20 µM for PINK-1 flies.

#### 4.3.3. Climbing Assay and Mortality Rate up to Two Weeks

For the motor activity, performed every 4 to 5 days, all groups were tested at random [63]. Groups of 10−15 flies (depending on the experiments) were transferred into an empty 100-mL Pyrex graduated cylinder with a foam plug, and a height of an 8-cm horizontal line above the bottom of the cylinder was drawn on a paper as a criteria. The flies were allowed 10 min to acclimatise. The flies were then gently tapped down and allowed to climb up past the 8-cm mark (in 8 s) on the chart and, afterwards, tapped down again. A digital camera was used to record the flies at a distance of 30 cm from the paper, and a timer was used to record the time. The total number of flies that crossed the 8-cm mark was recorded as the “escaped flies”. This was repeated twice more. The climbing assay was performed at 10 a.m. every two to three days. An average of the total number of flies that escaped was noted. Ability of surviving flies per day (%) was calculated by dividing the number of flies that climbed over the 8-cm mark by the total number of surviving flies multiplied by a hundred. The survival rate per day was calculated by dividing the number of surviving flies by the number of flies on day 0 (multiplied by a hundred). Experiment duration was for two weeks (from the start of the experiment). This was not a full lifespan experiment. The mean of each group was calculated using the data from the three replicates tests. An adjusted two-way multiple measures analysis of variance (ANOVA) with post-test Bonferroni correction was performed using GraphPad Prism, version 7 (GraphPad Software, San Diego, CA, USA).

#### 4.3.4. Biochemical Assays in Fly Brains

At the end of exposure to vanadium, the flies from each group of control and treated (vanadium only and vanadium + DFO) groups were anaesthetized on ice, snapped frozen in liquid nitrogen, and vortexed at high speed to separate the fly head from the body. The detached fly heads were transferred into pre-weighed Eppendorf tubes and weighed. They were then homogenised in 0.1-M phosphate buffer, pH 7.0 (1 mg: 10 µL), centrifuged for 10 min at 4000 g (4 °C). The supernatants obtained were stored at −20 °C and used to determine the ROS level, total thiol, and protein content. The assays were performed in duplicates (*n* = 3 replicates).

#### 4.3.5. Measurement of ROS in Drosophila Melanogaster WT and PINK-1 Mutant Fly Brains

To determine the ROS level following chronic exposure to vanadium and the influence of Fe chelation and 2′ and 7′-Dichlorofluorescein (DCFH), oxidation was measured as an index of oxidative stress using a 96-well plate [64]. The fluorescence product of DFH oxidation (i.e., DCF) was measured for 10 min (at 30-sec intervals) using a Synergy H4 hybrid multi-mode microplate reader (excitation set at 488 and 525 nm emission) (Tecan Trading AG, Switzerland). All the experiments were conducted in duplicates for each for *n* = 3 replicates. The rate of DCF formation was expressed in percentage of the control group.

#### 4.3.6. Total Thiol (T-SH) Assay in Drosophila Melanogaster WT and PINK-1 Mutant Fly Brains

The total thiol content for both WT and mutant flies was determined [65]. The reacting mixture contained 170 µL of 0.1-M potassium phosphate buffer (pH 7.4) and 20 µL of the sample, as well 10 µL of 10-mM DTNB. it was incubated for 30 min at room temperature; the absorbance was measured at 412 nm and used to calculate the sample total thiol levels (in µmol/mg protein) using GSH as the standard.

## Figures and Tables

**Figure 1 ijms-21-06719-f001:**
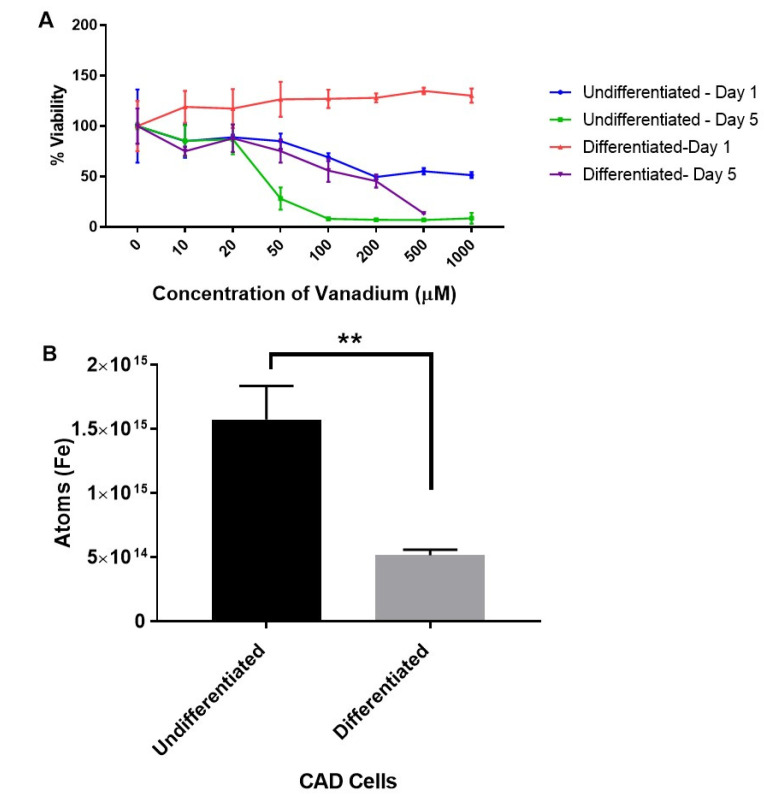
Dose response to vanadium on oxidative stress (mitochondrial viability) in undifferentiated and differentiated Catecholaminergic a-differentiated (CAD) cells (**A**), and the iron content in undifferentiated and differentiated CAD cells (**B**). Results from the mitochondrial viability investigation showed that undifferentiated neuronal cells (**A**) are more sensitive (ca. 10-fold) to vanadium than differentiated cells (**B**). This correlates with the significantly higher (*p* < 0.005) levels of intracellular iron in undifferentiated versus differentiated CAD cells. Values are mean ± SD for *n* = 4 replicates for the mitochondrial viability assay, and for the metal content analysis, values are mean ± SD from 2 separate experiments and *n* = 4 for each individual experiment (Student’s *t*-test ** *p* < 0.005).

**Figure 2 ijms-21-06719-f002:**
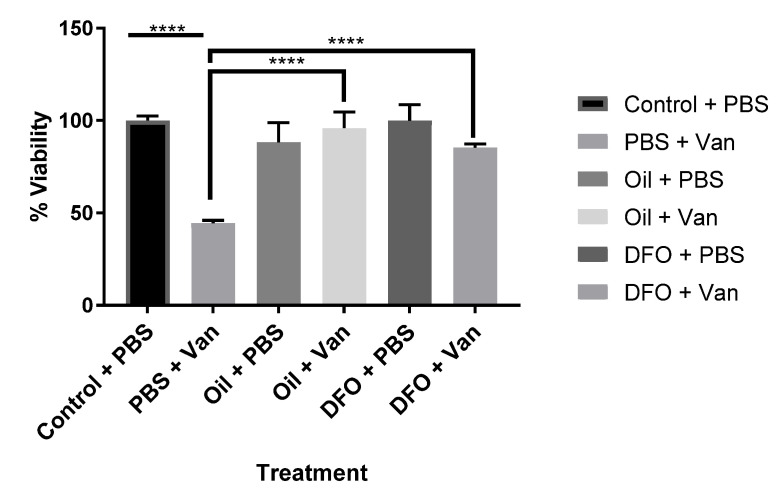
Effects of iron chelation on the chronic exposure to vanadium in differentiated CAD cells. The effects of the synthetic (deferoxamine, DFO) and natural iron chelator (*A. citrodora* essential oil) upon vanadium (200 μM)-induced oxidative stress (mitochondrial viability) following chronic (six-day) exposure for differentiating CAD cells. Both synthetic and natural chelators significantly reversed vanadium-induced toxicity, as compared with vanadium-only treated cells. The concentration of DFO used was 10 µM, and *A. citrodora* was 0.01 mg/mL. Values are mean ± SD for *n*= 4 samples. **** *p* < 0.0001. A repeated measure one-way ANOVA (with Tukey’s multiple comparisons test) was performed.

**Figure 3 ijms-21-06719-f003:**
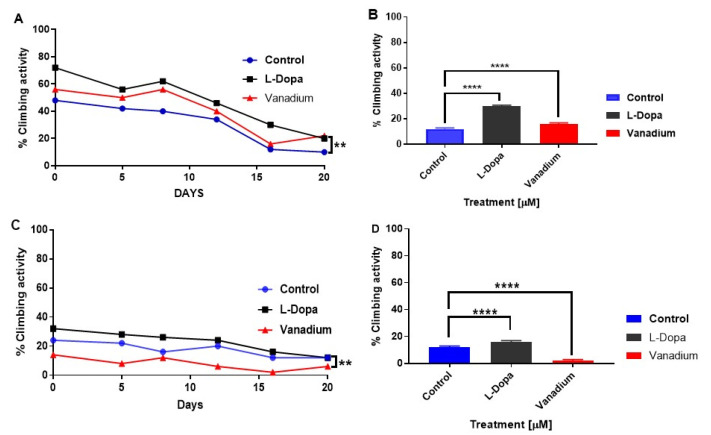
Effects of chronic exposure of vanadium on a percentage of flies that escaped in the climbing ability test for wild-type (WT) and PINK-1 mutant flies. In wild-type flies (**A**), a progressive decrease in the motor activity of the flies over time was observed across the three groups tested. A modest increase in motor activity can be seen with flies treated with L-dopa compared to the control group but no significant effect in the group treated with vanadium (*p* > 0.05). In contrast, PINK-1 flies (**C**) showed a highly significant lower motor activity than the WT flies and a progressive decrease in motor activity over time across the three groups. Notably, vanadium exacerbated the existing locomotor deficits in mutant PINK-1 flies (** *p* < 0.01). A repeated measure one-way ANOVA (with Tukey’s multiple comparisons test) was performed. (**B**,**D**) Show significant contrasting effects on the climbing ability after two weeks (day 16 for WT and mutant, respectively), analysed with a one-way ANOVA (Dunnett’s multiple comparisons test) was performed. (** *p* < 0.01, and **** *p* < 0.0001).

**Figure 4 ijms-21-06719-f004:**
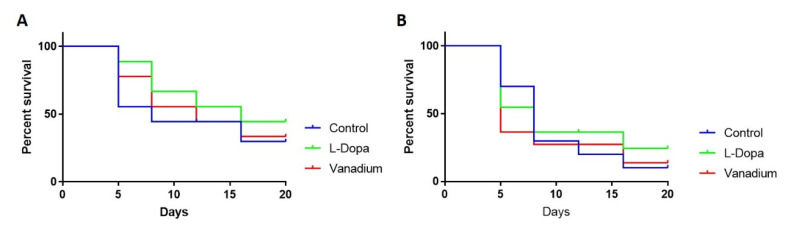
Effects of chronic exposure of vanadium on survival in WT (**A**) and PINK-1 mutant (**B**) flies. In both fly groups, a progressive decrease in the survival rate over time was observed across the three groups tested.

**Figure 5 ijms-21-06719-f005:**
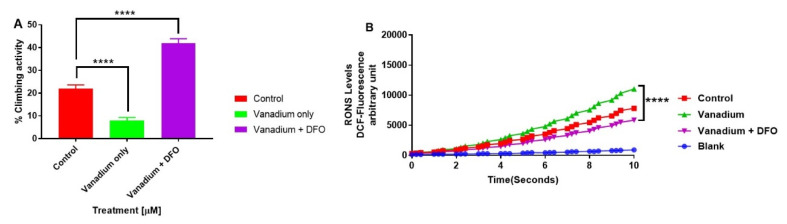
The influence of iron chelation following a chronic exposure of Drosophila melanogaster (PINK-1 mutant) to low-dose vanadium upon motor activity (**A**) and reactive oxygen species (ROS) production in fly brains (**B**) on day 14. Exposure of PINK-1 mutant to vanadium exacerbated the existing locomotor deficits in PINK-1 flies (**** *p* < 0.0001), comparing the slope of vanadium versus the control. However, a significant improvement was observed in the motor activity in the presence of the iron chelator (DFO) + vanadium (**** *p* < 0.0001), in comparison to vanadium alone. Results showed that the treatment with a low dose of vanadium enhanced ROS generation (**B**) after 14 days in PINK-1 mutant flies, when compared with the control group. In contrast, iron chelation (DFO) completely reversed the vanadium production of ROS (**B**), which is statistically significant (**** *p* < 0.0001) when compared with vanadium only. All values are means ± SD, from 5 separate replicates; *n* = 10 flies for each individual experiment; data were analysed with a repeated measure one-way ANOVA (with Tukey’s multiple comparisons test) (**** *p* < 0.0001).

**Figure 6 ijms-21-06719-f006:**
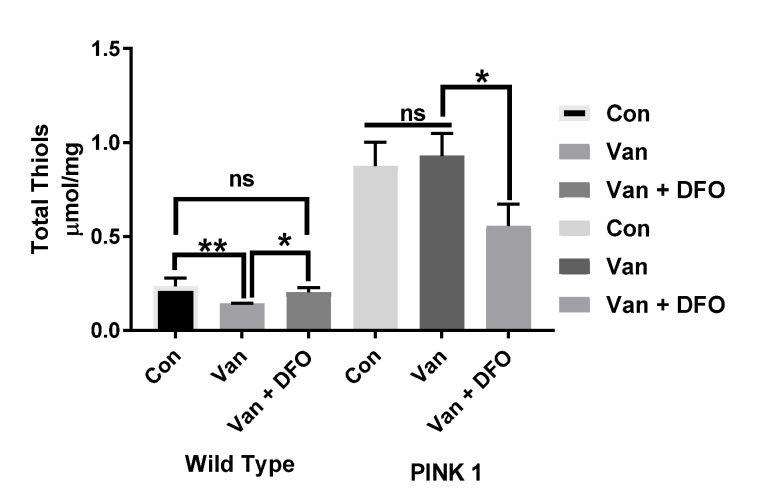
The influence of iron chelation following the chronic exposure of Drosophila melanogaster (WT and PINK-1 mutant) to low-dose vanadium upon the total thiol (T-SH) levels (after day 14). Influence of iron chelator T-SH levels in wild-type flies is lower than with those seen in PINK-1 mutant flies. Vanadium elicited a reduction in T-SH in WT flies but no significant effect upon PINK-1 flies. Iron chelation (DFO) reverses the effect of vanadium on T-SH levels in the WT fly to control levels (* *p* < 0.05) and significantly reduced the T-SH levels in PINK-1 flies (* *p* < 0.05) compared to the control and vanadium-treated PINK-1 flies. All values are from 5 separate experiments; *n* = 10 flies for each individual experiment, analysed with a repeated measure two-way ANOVA. * *p* < 0.05. ** *p* < 0.01, ns not significant The influence of iron chelation following the chronic exposure of Drosophila melanogaster (WT and PINK-1 mutant) to low-dose vanadium upon the total thiol (T-SH) levels (after day 14). Influence of iron chelator T-SH levels in wild-type flies is lower than with those seen in PINK-1 mutant flies. Vanadium elicited a reduction in T-SH in WT flies but no significant effect upon PINK-1 flies. Iron chelation (DFO) reverses the effect of vanadium on T-SH levels in the WT fly to control levels (* *p* < 0.05) and significantly reduced the T-SH levels in PINK-1 flies (* *p* < 0.05) compared to the control and vanadium-treated PINK-1 flies. All values are from 5 separate experiments; *n* = 10 flies for each individual experiment, analysed with a repeated measure two-way ANOVA. * *p* <0.05. ** *p* <0.01, ns not significant

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
