# Peer review of "The Pathopharmacological Interplay between Vanadium and Iron in Parkinson’s Disease Models"

_ijms, 2020, doi:10.3390/ijms21186719_

Round 1

Reviewer 1 Report

The manuscript entitled "THE PATHOPHARMACOLOGICAL INTERPLAY BETWEEN VANADIUM AND IRON IN PARKINSON’S DISEASE MODELS" by Ohiomokhare et al. studied the metal toxicity of vanadium (Vn) in multiple Parkinson's models, e.g. CAD cells and a Drosophila PINK1 mutant. They found in both models: iron excessively accumulated, as well as ROS. More importantly, two iron chelators, Aloysia and DFO can impede the toxicities and protect the disease models.

This study has potential interests to the reader in the field of metal toxicity related neurodegeneration, since many other heavy metals may also share a similar mechanism. The experiments are performed well too. However, I do have some major concerns to address:

First of all, how specific is DFO's chelating ability? The whole study relies on the rescue of DFO treatment. I think the genetic factors, such as ferritin and transferrin should be included to solid the author's findings. Otherwise, it could be possible that DFO directly removes the additive Vn from the medium, then we are chasing a wild goose here.

  1. In CAD cell studies, the authors measured iron levels. How about the ROS generation?
  2. Also in CAD cells, if providing extra Fe, will it make the differentiated cells more sensitive? If treating the cells with an iron-depleted medium, whether it will make the undifferentiated cell less sensitive?
  3. In figure 2, how about the iron levels of these groups?
  4. In fly models, the dPINK1 mutant usually shows abnormal wing postures and thorax indentation. Why the authors did not choose these phenotypes but the climbing index?
  5. In figure 4, will Fe+Vn give a stronger effect, and DFO can prevent it?
  6. The authors mentioned "Conversely, in the WT flies, treatment with low dose of VD significantly reduced ROS generation after 14 days compared to the controls and iron chelation (DFO) completely reverses this VD-induced reduction of ROS (not shown)." How should the authors interpret it?
  7. In figure 6, can we use anti-oxidants to rescue the Vn's toxicity? Also, can we use ferritin overexpression or SOD2 overexpression to rescue the Vn's effects? It would be very interesting if seeing Fe metabolic genes actually genetically interact with Vn's effects.

Author Response

The manuscript entitled "THE PATHOPHARMACOLOGICAL INTERPLAY BETWEEN VANADIUM AND IRON IN PARKINSON’S DISEASE MODELS" by Ohiomokhare et al. studied the metal toxicity of vanadium (Vn) in multiple Parkinson's models, e.g. CAD cells and a Drosophila PINK1 mutant. They found in both models: iron excessively accumulated, as well as ROS. More importantly, two iron chelators, Aloysia and DFO can impede the toxicities and protect the disease models.

Thank you to reviewer for positive comments.

First of all, how specific is DFO's chelating ability? The whole study relies on the rescue of DFO treatment. I think the genetic factors, such as ferritin and transferrin should be included to solid the author's findings. Otherwise, it could be possible that DFO directly removes the additive Vn from the medium, then we are chasing a wild goose here.

We agree with reviewer that the chelating activity and selectivity of DFO is a factor. We would disagree that the whole study depends on this. We have considered this and argued this point in the text. In fact we state that no chelator available is absolutely selective, in the text, that the DFO effect may be a combination chelation of iron/vanadium (lines 421-432), which still provided a viable candidate therapeutic strategy.

  1. In CAD cell studies, the authors measured iron levels. How about the ROS generation? We were interested in correlating endogenous iron content to sensitivity to oxidative stress-sensitivity. ROS, RONS etc will certainly be explored in future experiments.
  2. Also in CAD cells, if providing extra Fe, will it make the differentiated cells more sensitive? This is an interesting experiment, but we were interested in endogenous iron content and influence on exogenous vanadium, see below in vivo.  Also based on -Todorich et al (2011) observed that providing extra Fe via expression of transgenic human ferritin in rat primary oligodendrocyte progenitor cells exacerbated vanadium cytotoxicity. B. Todorich, J.O. Olopade, N. Surguladze, X. Zhang, E. Neely, J.R. Connor, The mechanism of vanadium-mediated developmental hypomyelination is related to destruction of oligodendrocyte progenitors through a relationship with ferritin and iron, Neurotox. Res. 19 (2011) 361–373. If treating the cells with an iron-depleted medium, whether it will make the undifferentiated cell less sensitive? Again an interesting future experiment, however, we showed DFO fully protected against oxidative stress induced mitochondrial viability in differentiated cells, closer to adult cell model more relevant to PD, and our proposal as a future therapeutic strategy
  3. In figure 2, how about the iron levels of these groups? Thank you for this suggestion. Again, a useful future experiment.
  4. In fly models, the dPINK1 mutant usually shows abnormal wing postures and thorax indentation. Why the authors did not choose these phenotypes but the climbing index? Yes, this model does shown various physical changes.  This is a Parkinson's disease model (focus of our work) and we are specifically interested in motor behavioural phenotype, and health and wellbeing, with relation to this disease.
  5. In figure 4, will Fe+Vn give a stronger effect, and DFO can prevent it? Possibly yes, but we, again, we wanted to explore in vivo endogenous iron and not add exogenous iron to experiment (iron is an endogenous heavy metal and vanadium is an exogenous environmental heavy metal. We wished to maintain this in our experimental design as much as possible (also in vitro, see above).
  6. The authors mentioned "Conversely, in the WT flies, treatment with low dose of VD significantly reduced ROS generation after 14 days compared to the controls and iron chelation (DFO) completely reverses this VD-induced reduction of ROS (not shown)." How should the authors interpret it? We interpret this with relation to the known, but still poorly understood, beneficial effects of vanadium in humans (eg. anti-oxidant, health supplement). Our study shows the fine balance between positive and negative effects of vanadium depending on the cellular environment (health ageing Vs disease)
  7. In figure 6, can we use anti-oxidants to rescue the Vn's toxicity? Also, can we use ferritin overexpression or SOD2 overexpression to rescue the Vn's effects? It would be very interesting if seeing Fe metabolic genes actually genetically interact with Vn's effects. Thank you for this excellent suggestion and we will certainly consider this going forwards.

Reviewer 2 Report

In this study, Ohiomokhare et al. address the toxicity of vanadium on Pink1 mutant Drosophila and catecholaminergic differentiated and undifferentiated cell lines as models of Parkinson’s disease. Vanadium exacerbates iron-related toxicity that may underlie the pathogenesis of Parkinson’s disease. They conclude that chronic vanadium exposure on a sensitized Pink1 mutant background makes flies more vulnerable to ROS accumulation and impairs their climbing ability and lifespan further. This parallels the findings in the catecholaminergic cell line where mitochondrial viability is compromised after vanadium treatment in an iron-dependent manner.

On the positive side, the concept of heavy metal accumulation with relation to Parkinson’s disease is getting attention and therapies based on iron chelation may be effective in combination with antioxidant treatment. The results clearly show on many occasions that the effect of vanadium is Pink1- and iron-dependent. That might open the scene for studies in more complex models, such as iPSC-derived differentiated, Parkinsonism-related mutation-harboring neurons and in vivomammalian models.

 However, there are still some shortcomings present in data collection and the interpretation of the results that would require moderate revision of the manuscript. Please see my specific comments below.

Major points

  1. Measurement of Drosophila lifespan is inadequate and requires additional experiments. From the Materials and Methods, I could only conclude that the same batch of flies was used for climbing tests and lifespan analysis. Prolonged tapping influences fly lifespan as evidenced by the shockingly short expected lifespan of wild-type flies without any treatment (Fig. 4A). As the experiment is not carried through until the natural death of the last survivor, this hardly constitutes a proper longevity assay in my opinion. Furthermore, flies should be counted and flipped every 2-3 days as it is also written in Materials and Methods with relation to climbing assays. However, Fig. 4 shows regularly 4-5 days between time points. Here, as the sex is not specified, I assume that this a mixed sex population where hatching larvae from eggs laid by females would result in sticky food that kills considerable number of flies after 5 days of incubation by trapping them. Please refer to appropriate methodological papers on fly lifespan analysis (such as Linford et al., doi:10.3791/50068 (2013).

  1. T-SH levels in wild-type flies decrease upon vanadium treatment that is rescued by DFO (Fig. 6). Nonetheless, the inherently high levels of T-SH in Pink1 mutants are unchanged after vanadium treatment and lowered by further DFO treatment. As the point to be made (if I assume well) is about the vanadium effect in Pink1 mutant flies - which is non-existent - this constitutes negative data. This comparison of vanadium treatment effects on wild-type and Pink1 mutants is not discussed in any detail anywhere in the text, only described. Please elaborate on the significance of this figure.

Minor points

  1. Line 21-22. PD-like phenotypes are evoked in the abstract referring to the cell culture results. However, only mitochondrial viability and iron content determination are described for cell lines. Mitochondrial viability may not be specific enough to be called a “PD-like phenotype”.
  2. Line 44. SNAc should be called SNCA
  3. Line 57. Ref. 13 only measures heavy metal accumulation in the olfactory system, not the basal ganglia as suggested
  4. 1A shows viability values at different V concentrations. These are connected by a continuous line. However, I expect that not the same cells were treated with the increasing concentrations of V and therefore it seems misleading to draw a continuous line. I suggest showing discrete measurements instead.
  5. Supplementary Figure 2 is not called out in the text

Author Response

  1. Measurement of Drosophila lifespan is inadequate and requires additional experiments. From the Materials and Methods, I could only conclude that the same batch of flies was used for climbing tests and lifespan analysis. Prolonged tapping influences fly lifespan as evidenced by the shockingly short expected lifespan of wild-type flies without any treatment (Fig. 4A). As the experiment is not carried through until the natural death of the last survivor, this hardly constitutes a proper longevity assay in my opinion. We agree with the reviewer that this was not a complete life span study. In fact, we were interested in exploring the effects upon health status of the ageing fly. Hence, the selection of 2 weeks. We selected this time point to simultaneously explore motor behaviour, oxidative status and health of aging fly. We have modified the text to reflect this point (pp187). Furthermore, flies should be counted and flipped every 2-3 days as it is also written in Materials and Methods with relation to climbing assays. However, Fig. 4 shows regularly 4-5 days between time points. Here, as the sex is not specified, I assume that this a mixed sex population where hatching larvae from eggs laid by females would result in sticky food that kills considerable number of flies after 5 days of incubation by trapping them. Yes, the 4-5 days were  assay time points, which does not necessarily correspond with fly flipping. Before experiment, both gender of flies were exposed to treatment (as indicated in the methods). However, Male flies were selected for the 2 week-long experiment. So the problem of egg hatching is eliminated. (Yes, the 4-5 days were  assay time points, which does not necessarily correspond with fly flipping (2-3 flipping days) (pp182, 187-188). Trauma is an issue in flipping and geotaxis assays can result in negative health  effects. However, all groups were treated in the same fashion, which controls for this confounder. Please refer to appropriate methodological papers on fly lifespan analysis (such as Linford et al., doi:10.3791/50068 (2013). Thank you to reviewer for this suggestion.

  1. T-SH levels in wild-type flies decrease upon vanadium treatment that is rescued by DFO (Fig. 6). This correlates with the positive effects upon behaviour. See other reviewer response. Nonetheless, the inherently high levels of T-SH in Pink1 mutants are unchanged after vanadium treatment and lowered by further DFO treatment. As the point to be made (if I assume well) is about the vanadium effect in Pink1 mutant flies - which is non-existent - this constitutes negative data. This comparison of vanadium treatment effects on wild-type and Pink1 mutants is not discussed in any detail anywhere in the text, only described. Please elaborate on the significance of this figure. We discuss the surprisingly high T-SH in PINK-1 flies in discussion. The lack of effect of vanadium, suggests that the T-SH levels are already maximal in PINK-1 fly. DFO partially reversed this maximal T-SH effect suggesting a partial iron-dependent mechanism present in PINK1 flies (pp398-399 and pp 407-408)

Minor points

  1. Line 21-22. PD-like phenotypes are evoked in the abstract referring to the cell culture results. However, only mitochondrial viability and iron content determination are described for cell lines. Mitochondrial viability may not be specific enough to be called a “PD-like phenotype”. This has been corrected in text
  2. Line 44. SNAc should be called SNCA This has been corrected
  3. Line 57. Ref. 13 only measures heavy metal accumulation in the olfactory system, not the basal ganglia as suggested. This has been corrected, "perhaps relevant to pre-motor anosmia symptom in PD" 
  4. 1A shows viability values at different V concentrations. These are connected by a continuous line. However, I expect that not the same cells were treated with the increasing concentrations of V and therefore it seems misleading to draw a continuous line. I suggest showing discrete measurements instead. This is a dose-dependent curve depicted the effects of of increasing concentration of Vanadium upon the cells, which as been standardised to 100% for each replicate experiment, so we feel this depiction is justified.
  5. Supplementary Figure 2 is not called out in the text Corrected pp313, 403

Round 2

Reviewer 2 Report

Thanks to the authors for the amendments and explanations. 

Two minor points left :

  1. SNCA should be used at line 44 (not corrected properly)
  2. line 409 - the effect of DFO on T-SH levels in Pink1 mutant necessarily indicates an iron-dependent mechanism while the authors present it in the manuscript as  "an iron-independent oxidative stress mechanism". In the response to reviewers, they nonetheless call this "a partial iron-dependent mechanism". Please resolve this discrepancy

Author Response

  1. SNCA should be used at line 44 (not corrected properly) Corrected
  2. line 409 - the effect of DFO on T-SH levels in Pink1 mutant necessarily indicates an iron-dependent mechanism while the authors present it in the manuscript as  "an iron-independent oxidative stress mechanism". In the response to reviewers, they nonetheless call this "a partial iron-dependent mechanism". Please resolve this discrepancy Corrected